# Effects of Cancer Presence and Therapy on the Platelet Proteome

**DOI:** 10.3390/ijms22158236

**Published:** 2021-07-30

**Authors:** Maudy Walraven, Siamack Sabrkhany, Jaco C. Knol, Henk Dekker, Inge de Reus, Sander R. Piersma, Thang V. Pham, Arjan W. Griffioen, Henk J. Broxterman, Mirjam Oude Egbrink, Henk M. W. Verheul, Connie R. Jimenez

**Affiliations:** 1Amsterdam UMC, Cancer Center Amsterdam, Vrije Universiteit Amsterdam, Medical Oncology, De Boelelaan 1117, 1081 HV Amsterdam, The Netherlands; maudywalraven@hotmail.com (M.W.); j.knol@amsterdamumc.nl (J.C.K.); henk.dekker@amsterdamumc.nl (H.D.); ingedereus@yahoo.com (I.d.R.); s.piersma@amsterdamumc.nl (S.R.P.); t.pham@amsterdamumc.nl (T.V.P.); a.griffioen@amsterdamumc.nl (A.W.G.); h.broxterman@amsterdamumc.nl (H.J.B.); 2Cardiovascular Research Institute Maastricht, Department of Physiology, Maastricht University Medical Center, 6229 ER Maastricht, The Netherlands; s.sabrkhany@maastrichtuniverisity.nl (S.S.); m.oudeegbrink@maastrichtuniversity.nl (M.O.E.)

**Keywords:** platelets, biomarkers, cancer, antitumor therapy, proteomics

## Abstract

Platelets are involved in tumor angiogenesis and cancer progression. Previous studies indicated that cancer could affect platelet content. In the current study, we investigated whether cancer-associated proteins can be discerned in the platelets of cancer patients, and whether antitumor treatment may affect the platelet proteome. Platelets were isolated from nine patients with different cancer types and ten healthy volunteers. From three patients, platelets were isolated before and after the start of antitumor treatment. Mass spectrometry-based proteomics of gel-fractionated platelet proteins were used to compare patients versus controls and before and after treatment initiation. A total of 4059 proteins were detected, of which 50 were significantly more abundant in patients, and 36 more in healthy volunteers. Eight of these proteins overlapped with our previous cancer platelet proteomics study. From these data, we selected potential biomarkers of cancer including six upregulated proteins (RNF213, CTSG, PGLYRP1, RPL8, S100A8, S100A9) and two downregulated proteins (GPX1, TNS1). Antitumor treatment resulted in increased levels of 432 proteins and decreased levels of 189 proteins. In conclusion, the platelet proteome may be affected in cancer patients and platelets are a potential source of cancer biomarkers. In addition, we found in a small group of patients that anticancer treatment significantly changes the platelet proteome.

## 1. Introduction

Survival of most patients with cancer is dependent on early discovery of the malignancy and treatment at an early stage [1]. Blood biomarkers of cancer could assist physicians in discovery of the disease, and in making clinical decisions throughout antitumor treatment. Until now, only a few clinically relevant cancer biomarkers have been discovered [2,3,4]. Thus far, most researchers have focused on biomarkers in serum or plasma of patients with regard to the presence or absence of cancer or the effect of therapy. We proposed that platelets, which contain vast amounts of proteins, could serve as a new source in the search for cancer biomarkers [5,6].

Platelets play an important role in primary hemostasis [7], inflammation [8], (tumor) angiogenesis [9,10] and cancer metastasis [11]. In addition, there are several studies which indicate that platelets could be of importance in the search for biomarkers of early-stage cancer [12]. Platelets, which are produced by megakaryocytes in the bone marrow, contain numerous proteins that bring about change (e.g., growth factors, chemokines and proteases). These proteins are either synthesized by the parental megakaryocytes or absorbed from the (micro)circulation [9,13,14]. Besides proteins, platelets are also able to sequester drugs such as bevacizumab [15] and sunitinib [16]. In addition, upon activation, platelets are able to secrete their diverse content [17], which enables affecting (tumor) angiogenesis [9,10,18,19], cancer cell proliferation [20] and migration [21] and metastasis [22,23].

Studies in mice demonstrate that the presence of a human tumor xenograft affects platelet proteins with angiogenic or angiostatic functions, whilst their plasma levels did not change [14,24,25]. Therefore, it was postulated that changes in platelet protein content might indicate the presence or recurrence of a malignancy. While numerous studies in cancer patients identified thrombocytosis as a predictor of poor prognosis [26], human studies investigating the effect of tumor presence on platelet characteristics and platelet content are rare. We demonstrated that several platelet characteristics, as well as the platelet proteome, are changed in patients with early-stage lung or pancreatic cancer [27,28]. Furthermore, we showed that a combination of these changes could be exploited to discriminate patients with early-stage disease from healthy sex- and age-matched individuals.

The aim of our current study was (1) to identify potential biomarkers of cancer that could be used to discriminate cancer patients from healthy individuals, and (2) to explore the effect of cancer therapy on the platelet proteome. We included patients with a variety of cancer types and healthy controls. Using global protein profiling by mass spectrometry-based proteomics, we demonstrated that the platelet proteome is not only affected by cancer presence, but may also be affected by antitumor therapy.

## 2. Results

### 2.1. Comparison of Protein Content of Platelets of Cancer Patients versus Healthy Individuals

The platelet protein content of patients with cancer was compared to healthy volunteers to investigate whether cancer-associated proteins could be distilled. Prior to therapy or after progressive disease following prior antitumor treatment, platelets from nine patients with different tumor types were isolated, washed and processed for proteomics (Figure 1). The tumor types included colon, rectal, anal, esophageal, gastric, tongue and primitive neuroectodermal cancer.

In addition, from ten healthy volunteers, platelets were isolated and processed at the same time. Additionally, from three patients, a second sample was collected during therapy to study the effect of antitumor therapy on the platelet proteome. Table 1 shows the characteristics of the patients and healthy controls. The median ages of patients and controls were 61 and 52 years, respectively. The comedication of the volunteers was not available.

Platelet lysate samples were pre-fractionated by gel electrophoresis (Appendix A). The proteins were in-gel digested with trypsin and analyzed on a nanoLC-MS/MS platform including a Q Exactive orbitrap mass spectrometer. Database searching identified 4200 protein groups linked to 4059 unique proteins (Appendix A), with an average of 2912 identified proteins in patient samples and 2808 in healthy control samples. As a quantitative measure for protein abundance, we used spectral counting [29]. One hundred and eighteen unique proteins were significantly different in abundance (*p* < 0.05) in samples of patients with cancer compared to healthy controls (Appendix A).

Unsupervised hierarchical cluster analysis and principal component analysis using normalized spectral count data do not show clear clustering of samples (Appendix A), whereas hierarchical clustering using differential proteins did show a clear separation of cancer patients from all but one healthy control (Figure 2).

Fifty differential proteins were more than 1.5-fold more abundant in patients (Appendix A), and 36 more in healthy volunteers, respectively (Appendix A). Twenty proteins were exclusively found in patients. Due to the heterogeneity of cancer types and the low protein abundancies, these proteins were not detected in all patients.

### 2.2. Functional Analysis of Differential Proteins in Platelets of Cancer Patients and Healthy Volunteers

To analyze the biological functions associated with the proteins that were significantly different in abundance, we focused on the 50 and 36 proteins that were at least 1.5-fold higher in abundance in patients and controls, respectively. Protein networks were created using protein–protein association data from the STRING database [30], and visualized in Cytoscape (Figure 3). In addition, gene ontology mining was performed using the BiNGO Cytoscape app to identify underlying biological processes. Proteins with higher abundance in cancer were mostly associated with inflammatory and immune responses (Figure 3A), while in healthy controls these were mostly involved in amino acid metabolism (Figure 3B).

The networks were generated using default settings in String and visualized using Cytoscape.

### 2.3. Antitumor Treatment May Affect the Platelet Proteome of Cancer Patients

From three patients, platelet samples were collected both before and several weeks after initiation of antitumor treatment. The protein content appeared to be considerably different in samples collected on-treatment. A distinct difference in the band pattern on the gel was observed compared to pre-treatment (Appendix A). In addition, hierarchical cluster analysis using differential proteins (paired statistics, *p* < 0.05) showed a clear clustering of before- and on-treatment samples, respectively (Figure 4).

Antitumor therapy led to a significant change in expression of 713 platelet proteins, with treatment leading to upregulation or downregulation of 432 and 189 proteins (>1.5-fold), respectively (Appendix A). The size of the major band observed on gel in on-treatment samples is in line with that of HBB (hemoglobin beta chain, 16 kDa, *p* = 0.006) and HBA1 (hemoglobin alpha 1, 15.3 kDa, *p* = 0.006). One hundred and fifty-five proteins were uniquely identified on-treatment (Appendix A), and 35 only pre-treatment (Appendix A). Ten of the 621 proteins that were differential in the treatment setting were among the 86 proteins that were differential in the comparison of patients versus healthy volunteers.

The proteins being significantly higher in abundance after the start of antitumor treatment (fold change > 5) (Figure 5) were generally linked to protein/RNA synthesis and transport, RNA splicing and processing, DNA repair and telomere maintenance, gas/drug transport and erythrocyte homeostasis, regulation of apoptosis, myeloid cell activation and regulation of catabolism.

The proteins with a higher abundance before treatment compared to on-treatment (Appendix A) were linked mostly to mitochondrial organization and cellular respiration. Combined, the above data show that antitumor treatment may change the platelet proteome of patients.

### 2.4. Selection of Potential Platelet Biomarkers of Cancer

To find cancer-associated proteins in platelets, our data were compared to those from a previously published paper from our group, in which the platelet proteome of patients with early-stage lung or pancreatic cancer (*n* = 12) was compared to platelets of a healthy sex- and age-matched control group (*n* = 11) [28]. The results of both studies displayed many similarities, including a comparable number of identified unique proteins (4384 versus 4059) and a large overlap between the protein sets (72%). Additionally, the number of platelet proteins that were significantly differential in cancer patients versus healthy controls was similar in both studies (104 versus 118). In order to find potential biomarkers of cancer, proteins that were significantly differential (>1.5-fold, *p* < 0.05) in both studies were selected (Figure 6, Table 2).

Six of these proteins had a higher abundance in cancer patients than in healthy controls (RNF213/ring finger protein 213; CTSG/cathepsin G; PGLYRP1/peptidoglycan recognition protein 1; RPL8/ribosomal protein L8; S100A8/S100 calcium binding protein A8; S100A9/S100 calcium binding protein A9). Two proteins were found at higher levels in healthy controls (GPX1/glutathione peroxidase 1; TNS1/tensin 1). Two additional proteins, AMDHD2/amidohydrolase domain-containing 2 and ERAP1/endoplasmic reticulum aminopeptidase 1, were significantly differential in both studies, but with opposite directions of change. Therefore, RNF213, CTSG, PGLYRP1, RPL8, S100A8, S100A9, GPX1 and TNS1 were identified as promising cancer-associated proteins which may provide non-invasive cancer biomarkers.

## 3. Discussion

In the current study, we demonstrate that the platelet proteome of patients may be affected by cancer presence and cancer therapy. Using these data in combination with a previous cancer platelet proteomics dataset [28], we defined eight platelet proteins (RNF213, CTSG, PGLYRP1, RPL8, S100A8, S100A9, GPX1 and TNS1) as promising cancer-associated proteins that may provide, upon further validation, cancer biomarkers within blood platelets.

The presence of an active malignancy influences multiple platelet parameters [27,28,31,32]. We previously showed that several platelet characteristics (e.g., count, volume, activation status, and angiogenic/angiostatic content) are affected in early stages of cancer [27]. In addition, a combination of these platelet features could be used to discriminate patients with early-stage lung or pancreatic cancer from healthy sex- and age-matched controls. In another independent study, the prospect of platelets as a potential source of cancer biomarkers was also highlighted. In the latter study, it was demonstrated that changes in platelet mRNA profiles could also be exploited to distinguish cancer patients from healthy individuals [33,34].

While proteomics is often used in the search for potential cancer biomarkers in plasma or serum, thus far, the platelet proteome has been largely overlooked, despite recent applications of proteomics in other diseases [35,36]. A proteomics study in tumor-bearing mice demonstrated that platelets are able to sequester tumor-derived proteins from clinically undetectable tumors [14]. Furthermore, we recently demonstrated for the first time that the platelet proteome of cancer patients is already different from that of healthy individuals in early stages of the disease, and normalizes after surgical resection of the tumor, thus presenting a lucrative source for potential cancer biomarkers [28].

In the current study, we explored whether cancer-associated proteins could be identified in an independent group of nine patients with various types of malignancies (e.g., colon, rectal, anal, esophageal, gastric, tongue and primitive neuroectodermal cancer). We found substantial changes in the platelet proteome of these patients compared to healthy controls. We identified a total of 4059 unique proteins in platelets. Amongst these proteins, 50 proteins were found at levels that were more than 1.5-fold higher in cancer patients, while 36 proteins were more than 1.5-fold higher in healthy volunteers. These numbers are highly comparable to data from our previous study in which we identified 4384 non-redundant proteins, of which 61 were more abundant in platelets of patients with early-stage lung or pancreatic cancer, and 24 proteins being more abundant in platelets of healthy controls [28]. The proteins that had markedly increased levels in platelets of patients are mostly associated with inflammatory and immune responses. We realize that the differences in median age of patients and controls and the use of comedication, as well as the various tumor types in patients, might influence protein content of platelets. Therefore, one should be careful with drawing conclusions, especially due to the small number of patients. The current number of patients is too small to study a correlation with tumor type. We did, however, see a potential tumor-type dependent change in the platelet proteome in our previous study (pancreatic versus lung cancer) [28].

These results are in agreement with previous studies performed in animals and humans which demonstrate that a malignant tumor triggers such responses in the host body [37]. Proteins that are more abundant in platelets of patients are produced or taken up by megakaryocytes from which they are derived, or taken up from circulation by platelets themselves. Megakaryocytes [38] and platelets [14] are both able to sequester (tumor-derived) proteins from their microenvironment which could affect their proteome. Megakaryocyte functions can also be influenced or hijacked by cytokines and interleukins that are produced by tumors, affecting megakaryopoiesis and platelet content [39,40].

In the present study, we also explored the effect of antitumor treatment on the platelet proteome of patients, which profoundly changed after initiation of treatment. Analysis of matched samples of three patients, collected before and after drug administration, revealed 432 proteins to be more abundant and 189 proteins to be less abundant after starting intervention. The proteins exhibiting higher levels on-treatment were linked to anabolic functions that are in line with a (re)active state of the megakaryocyte/platelet compartment, as well as regulation of apoptosis. Proteins that were more abundant pre-treatment were associated with mitochondrial organization and function. It is known that most cytotoxic drugs exert their effect by inducing apoptosis, immunological cell death, or other non-apoptotic cell deaths such as mitotic catastrophe, senescence and autophagy [41,42,43]. As a consequence, the cellular and/or host response is to increase processes such as protein translation, transport and biosynthesis [44,45]. Furthermore, previous studies have shown that cancer therapy can affect mitochondrial function [46,47,48,49]. In addition, it was demonstrated that platelet dysfunction in thrombocytopenic patients who were undergoing chemotherapy was due to impaired mitochondrial functioning [50]. In our study, the protein profiles within platelets appear to reflect the effect of cancer therapy. These alterations could be due to a direct effect of the therapy on platelets. However, it could also reflect the response of the body and/or the tumor to the treatment. We are aware of the fact that it is difficult to draw thorough conclusions from this small number of patients consisting of a variety of tumor types and receiving different antitumor treatment regimens. More extensive research is therefore required.

Based on the comparison of our current data with our prior platelet proteomics study [28], we identified eight cancer-associated proteins within platelets. In both studies, six proteins were detected at levels that were more than 1.5-fold higher in platelets of cancer patients than in healthy controls, and two proteins had more than 1.5-fold lower levels. The pertinent proteins are known to be involved in cancer development and progression. RNF213 encodes an E3 ubiquitin-protein ligase that is known to degrade NFAT1, a transcription factor that can in turn activate MDM2 to promote p53 degradation [51,52]. RNF213 is also important in tumor survival in hypoxic environments, and has been shown to be mutated in several cancer types [53,54,55]. CTSG is a potent platelet activator [56] and an endoprotease which has a role in cell migration, in eliminating intracellular pathogens, and in the breakdown of tissues at inflammatory sites [57]. CTSG is also highly expressed in various cancer types and is associated with tumor angiogenesis and metastasis [57,58,59]. PGLYRP1 is part of the innate immune system and has a direct antibacterial effect. It can also induce cytotoxic activity against tumor cells by activating NK cells and T lymphocytes [60]. S100A8 and S100A9 are calcium-binding proteins that normally occur as heterodimers [61] known as calprotectin [62]. They are known for their role in inflammatory responses and cancer [63,64] and are proposed as potential cancer biomarkers [65,66,67]. RPL8, a ribosomal protein involved in protein synthesis, is significantly overexpressed in various types of tumor cells and is a predictor of clinical outcomes in patients with cancer [68]. In contrast to the above six proteins, GPX1 and TNS1 were both found at a lower abundancy in platelets of cancer patients. GPX1 is one of the most important antioxidant enzymes in humans. Loss of its expression is reported in tumor samples of cancer patients, and downregulation appears to correlate with poor prognosis [69,70,71,72]. TNS1 is involved in cell adhesion, binding actin filaments and regulating actin polymerization. It is also important in malignant diseases, enhancing tumor suppressor RhoA [73]. Interestingly, three proteins of the proteins found in this study (RNF213, RPL8, GPX1) were also discovered as differentially expressed in a study in which platelet mRNA of cancer patients was compared to that of healthy individuals [33]. In future research, studies are warranted in larger numbers of patients to validate our results in order to translate these to the clinics. In addition, it will be of interest to study whether changes in platelets could be exploited in a predictive setting to distinguish responders from non-responders. Other clinical decision parameters, including the risk of treatment-related complications, such as bleeding or thrombosis, could possibly be substantiated on the basis of platelet proteome profiles as well.

In conclusion, the current study shows that the platelet proteome may be affected by cancer presence and antitumor therapy. In addition, eight cancer-associated proteins within platelets were defined that were also identified in our previous study. Future research is needed to further validate our current findings and to investigate whether platelets can be used as a repository of diagnostic and/or predictive cancer biomarkers. This study highlights the potential of blood platelets as a rich source of information that should be explored in further biomarker research.

## 4. Materials and Methods

### 4.1. Healthy Volunteers and Patients

In accordance with the declaration of Helsinki, study approval was obtained from the institutional review board of Amsterdam UMC. After informed consent was obtained, ten milliliters of blood was collected from ten healthy volunteers and nine cancer patients.

The patients that were selected were newly diagnosed with cancer and prior to the start of antitumor treatment, or had progressive disease following prior antitumor treatment. Subject details are shown in Table 1. From three patients, a second blood sample was collected at least one month after the start of antitumor treatment consisting of chemotherapy, or chemotherapy in combination with surgery and/or radiotherapy.

### 4.2. Platelet Isolation and Lysis

A schematic overview of the workflow is shown in Figure 1. Freshly drawn blood was collected by free-flow in 1/10 volume of 130 mM trisodium citrate. Platelet-rich plasma (PRP) was prepared by centrifugation (156× *g*, 15 min, 20 °C). To prevent platelet activation during further centrifugation, the pH of PRP was lowered to pH 6.5 by addition of 0.1 volume ACD (2.5% trisodium citrate, 1.5% citric acid, 2% D-glucose). A platelet pellet was obtained by centrifugation (330× *g*, 15 min, 20 °C). Next, the pellet was resuspended in Hepes-Tyrode (HT) buffer (145 mM NaCl, 5 mM KCl, 0.5 mM Na_2_HPO_4_, 1 mM MgSO_4_, 10 mM Hepes, 5 mM D-glucose, pH 6.5) in approximately the same volume as the collected PRP. Prostacyclin (PGI2, Cayman Chemical, Ann Arbor, MI, USA) was added to a final concentration of 10 ng/mL to prevent aggregation. A platelet pellet was obtained by centrifugation (330× *g*, 15 min, 20 °C). This washing step was repeated once. Platelets were resuspended in HT buffer pH 7.3 (2 × 10^11^ platelets per liter). The tubes were then centrifuged as above, in the presence of PGI2. Finally, the platelet pellet was resuspended in 100 µL sample buffer, heated for 5 min at 99 °C, and stored at −20 °C until further analysis.

### 4.3. Gel Electrophoresis and In-Gel Digestion

Precast 4–12% polyacrylamide gradient gels (NuPAGE, Fisher Scientific, Landsmeer, The Netherlands) were used for all samples. Samples were corrected for platelet count and, if necessary, diluted in sample buffer before loading a total of 25 µL on the gel. Proteins were separated at 125 V and stained with Coomassie Brilliant Blue G-250 (Fisher Scientific, Landsmeer, The Netherlands), washed in MilliQ water and stored at 4 °C until processing by in-gel digestion.

Based on our benchmark study [74] and other clinical proteome studies including platelet protemics of the past decade [28,75,76], we used in-gel digestion coupled to nanoLC-MS/MS analysis and spectral counting as a robust and reproducible proteomics workflow. After protein fractionation by gel electrophoresis, each lane on the gel was cut into ten gel slices. The gel slices were then cut into small cubes, and washed and dehydrated once in 50 mM ammonium bicarbonate (ABC) and twice in ABC containing 50% acetonitrile (ACN). Subsequently, gel cubes were incubated for one hour at 56 °C in ABC containing 10 mM dithiothreitol to break cysteine bonds, and for 45 min at room temperature (in the dark) in ABC containing 50 mM iodoacetamide to alkylate cysteines. The cubes were subsequently washed with ABC and ABC/50% ACN, dried in a vacuum centrifuge and incubated overnight at 23 °C with 6.25 ng/mL sequencing-grade modified trypsin (Promega, Leiden, The Netherlands). Extraction of peptides was performed once in 1% formic acid and twice in 5% formic acid/50% ACN. Prior to LC-MS/MS analysis, ACN was evaporated, and the extract volume reduced to 50 µL, in a vacuum centrifuge.

### 4.4. NanoLC-MS/MS

LC-MS/MS analysis was performed with an Ultimate 3000 Nano LC system (Dionex LC-Packings, Amsterdam, The Netherlands) coupled online to a Q Exactive mass spectrometer (Thermo Fisher, Bremen, Germany). See also Piersma et al. [17]

NanoLC. The binary buffer system consisted of buffer A (0.5% acetic acid), and buffer B (80% acetonitrile, 0.5% acetic acid). Following sample injection by the autosampler, peptides were trapped on a 10 mm, 100 μm ID trap column (5 μm, 120 Å ReproSil Pur C18aqua particles: (Dr Maisch GMBH, Ammerbuch-Entringen, Germany); 2% buffer B; 6 μL/min) and separated in 90 min over a 20-cm, 75-μm ID column (3 μm, 120 Å ReproSil Pur C18 aqua particles; 10–40% buffer B gradent; 300 nL/min). The inject-to-inject cycle was 120 min.

MS/MS. Peptides eluting from the column were ionized at 2 kV and sprayed into the mass spectrometer. Full survey MS scans in the orbitrap of the apparatus recorded accurate masses and intensities of intact peptides (resolution 70,000 at *m*/*z* 200, Automatic Gain Control target value of 3 × 10^6^ charges). The top 10 peptide ions with the highest intensity (excluding singly charged ions) were selected for isolation and fragmentation in the Higher-energy Collissional Dissociation cell of the apparatus (4 amu isolation width, 25% normalized collision energy). MS/MS spectra of the resulting fragments were recorded in the orbitrap (resolution 17,500 at *m/z* 200, AGC target value of 2 × 10^5^ charges, underfill ratio of 0.1%). For top 10 selection of peptides, a dynamic exclusion rule was applied with a repeat count of 1 and an exclusion time of 30 s.

### 4.5. Protein Identification

Database searching was performed with MaxQuant version 1.4.1.2. as described in Piersma et al. [17], using a Uniprot human reference proteome FASTA file (2014_01 with fragments removed, 61,552 entries). Searches were made specific for trypsin, allowing two missed cleavages, as well as methionine oxidation and N-terminal acetylation as variable modifications. Cysteine carboxamidomethylation resulting from iodoacetamide treatment of samples was required as a fixed modification. Searching used mass deviation windows of 4.5 ppm and 20 ppm for MS (intact peptides) and MS/MS (fragmented peptides), respectively. We required a minimal peptide length of 7 amino acids, a minimal Andromeda score for modified peptides of 40, and a minimum delta score (with respect to the second hit) of 6. Using a decoy database strategy, the false discovery rate for both peptide and protein identification was set at 1%. Multiple protein candidates that could not be discriminated on the basis of MS/MS evidence were combined into protein groups. Quantification was label-free. The mass spectrometry data have been submitted to the ProteomeXchange Consortium via the PRIDE partner repository [77] with the dataset identifier PXD010380.

### 4.6. Data Analysis

Proteins were quantified by spectral counting, that is, the number of identified MS/MS spectra for a given protein [29]. Raw counts were normalized on the sum of spectral counts for all identified proteins in a particular sample, relative to the average sample sum determined with all samples. Zero values were treated as protein expression values and not imputed. Beta-binominal statistics were used to assess changes in protein levels in platelet proteomes of healthy volunteers versus patients with cancer [78]. Paired statistics was used to compare the pretreatment samples to matched samples collected after the start of antitumor treatment [79]. The statistical tests were designed to take into account technical variation of spectral count data and implemented in the R package count data. Additional general protein information was retrieved by Ingenuity Pathway Analysis (Ingenuity^®^ Systems, www.ingenuity.com). Protein–protein associations were retrieved from the STRING database (www.string-dg.org, version 10) [30] and loaded as a network into Cytoscape (www.cytoscape.org) [80]. Highly connected protein clusters were identified with the Cytoscape plug-in ClusterONE [81] and gene ontology analysis was performed with the BINGO plug-in [82].

## Figures and Tables

**Figure 1 ijms-22-08236-f001:**
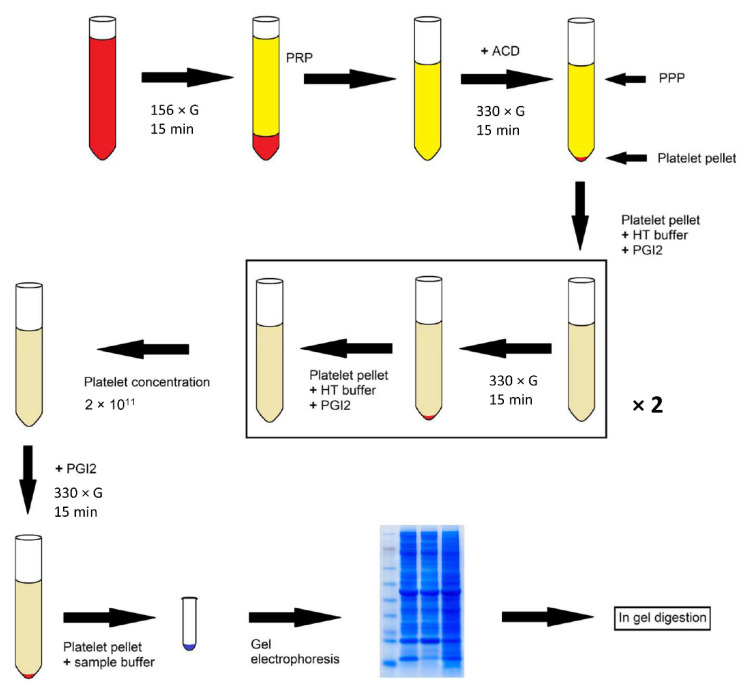
Schematic representation of the workflow. Blood was collected in citrated tubes. Platelet-rich plasma (PRP) was prepared by centrifugation. ACD was added, after which a platelet pellet was obtained by centrifugation. The pellet was resuspended in Hepes-Tyrode (HT) buffer (pH 6.5). Prostacyclin (PGI2) was added and a pellet was obtained by centrifugation. This was repeated once. Platelets were resuspended in HT buffer (pH 7.3). Tubes were centrifuged in the presence of PGI2. The pellet was resuspended in sample buffer, heated for 5 min at 99 °C and stored at −20 °C. Proteins were separated with electrophoresis (125 V) in precast gradient gels. In-gel digestion was performed.

**Figure 2 ijms-22-08236-f002:**
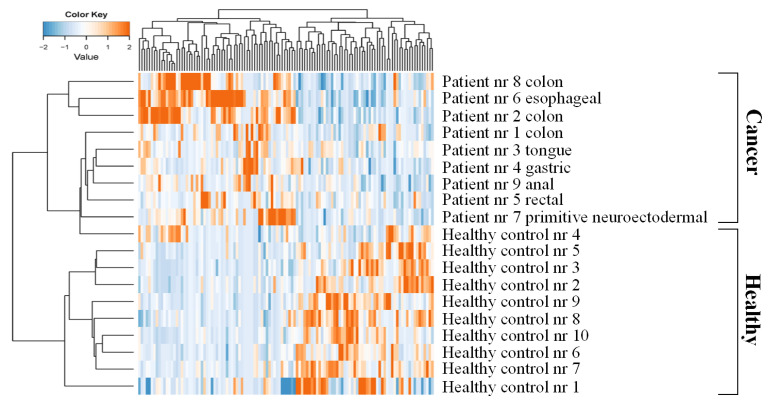
Platelet proteome profiling. Hierarchical clustering using proteins with differential abundance between platelets of patients with cancer (*n* = 9) and healthy controls (*n* = 10).

**Figure 3 ijms-22-08236-f003:**
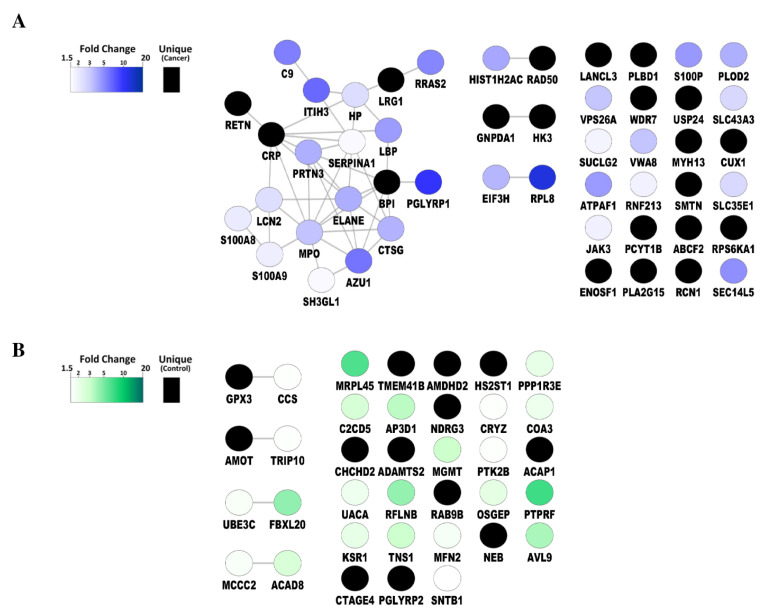
Protein–protein networks of significantly upregulated platelet proteins (**A**) in patients with cancer compared to healthy volunteers, (**B**) in healthy volunteers compared to patients with cancer. (*p* < 0.05, fold change ≥ 1.5).

**Figure 4 ijms-22-08236-f004:**
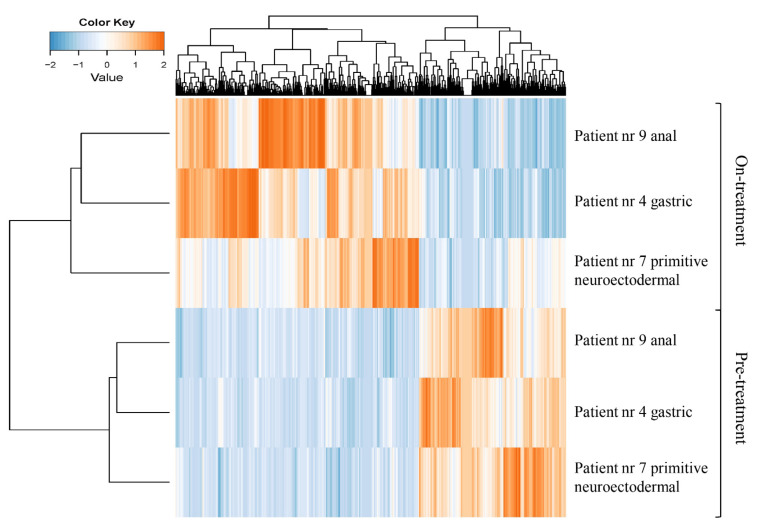
The platelet proteome may be affected by cancer treatment. Heat map and supervised clustering using proteins with differential abundance between platelets of patients with cancer (*n* = 3) before the start of antitumor treatment compared to paired platelets samples after initiation of antitumor treatment.

**Figure 5 ijms-22-08236-f005:**
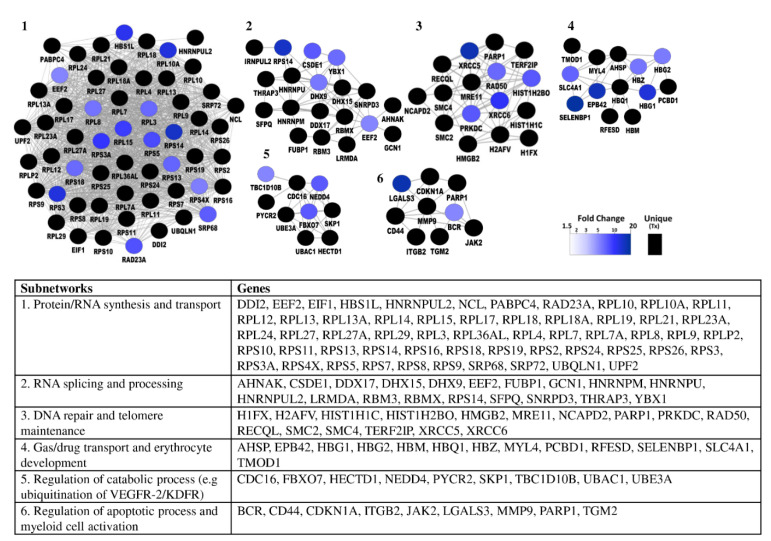
Protein–protein networks of significantly upregulated platelet proteins after the start of antitumor treatment. Platelet protein content was compared in paired samples of patients with cancer (*n* = 3) which were drawn pretreatment and after initiation of antitumor treatment. (*p* < 0.05, fold change ≥ 5). The networks were generated with cytoscape app “ClusterONE”.

**Figure 6 ijms-22-08236-f006:**
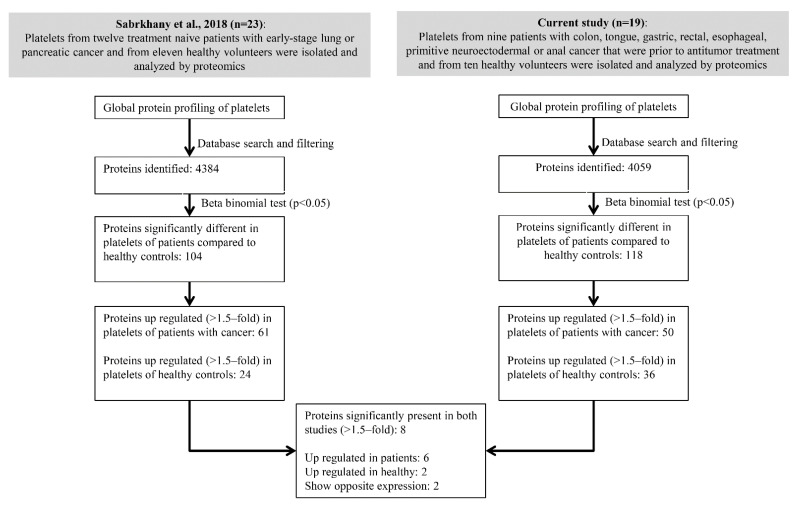
Flowchart depicting the steps used to select the most promising cancer-associated proteins within platelets, of which eight overlapped with our previous cancer platelet proteomics study [28].

**Table 1 ijms-22-08236-t001:** Characteristics of the patients and healthy volunteers. Radiofrequency ablation (RFA), radiotherapy (RT), carpal tunnel syndrome (CTS), (paroxysmal) atrial fibrillation ((P)AF), benign prostatic hyperplasia (BPH), coronary artery bypass graft (CABG), percutaneous transluminal coronary angioplasty (PTCA) (* platelet counts one day before or after collection, ** platelet counts 2.5 months before and 1.5 month after sample collection).

Healthy Volunteer	Gender	Age	Patient	Gender	Age	Tumor Type	Stage	Comorbidity/Previous Treatment	Medication	Platelets (×10⁹) 1st Sample	Treatment within 2 Samples	Platelets (×10⁹) 2nd Sample
1	M	58	1	F	79	Colon Cancer	IV	Prior treatment colon cancer (surgery, RFA, irinotecan, oxaliplatin, fluorouracil, capecitabine, pemetrexed), Cholecystectomy, Hypertension, Varicosis, Angina pectoris, Lung embolism, Hernia Inguinalis, Bladder correction, Arthrosis, Hepatitis A	Acetylsalicylic acid, Isosorbide mononitrate, Amlodipine, Lansoprazole, Candesartan, Macrogol	273/310 **		–
2	M	47	2	M	77	Colon Cancer	IV	Prior treatment colon cancer (surgery, RFA, capecitabine, oxaliplatin), PAF, Aortic valve insufficiency, Hypertension, Herpes zoster infection, Ulcus duodeni, Benign prostate hyperplasia, Pyelourethral stenosis left kidney, Hernia inguinalis	Alfusozine	518		–
3	M	42	3	M	56	Tongue Cancer	IV	Arthrosis spinal vertebrae	No medication	394		–
4	M	58	4	M	56	Gastric Cancer	IV	Primairy Immunodeficiency Syndrome (hypogammaglobuliaemia), Celiac disease, Osteoporosis	Immunoglobulines, Venofer	174	Gemcitabine, Cisplatine	118
5	M	51	5	M	61	Rectal Cancer	IV	Prior treatment rectal cancer (surgery, capecitabine, oxaliplatin, irinotecan, bevacizumab, radiotherapy)	Magnesiumhydroxide	260		–
6	M	60	6	M	70	Esophageal Cancer	IV	Prior treatment esophageal cancer (epirubicine, capecitabine, cisplatine, surgery), Appendectomy, Abdominal aortic aneurysm surgery	Oxycontin, Fentanyl, Paracetamol, Omeprazole, Temazepam	259 *		–
7	M	59	7	M	50	Primitive Neuroectoder-mal Cancer	III	Nose septum correction, Meniscus surgery, AF	Acetylsalicylic acid and Metoprolol (both started before sample 2 but after collection of sample 1)	335 *	Etoposide, Vincristine, Doxorubicine, Ifosfamide, Actinomycine-D	147
8	F	52	8	M	51	Colon Cancer	IV	Prior treatment colon cancer (surgery, RFA, bevacziumab, capecitabine, irinotecan, fluorouracil, oxaliplatin)	Acetylsalicylic acid, Metoprolol, Metformin, Oxycontin, Pantoprazole, Diclofenac (if necessary)	339		–
9	M	50	9	M	67	Anal Cancer	III	CABG, cardiac valve prothesis, PTCA	Acetylsalicylic acid, Plavix, Simvastatin, Lisinopril, Paracetamol/tramadol	196	Fluorouracil, Mitomycine C, Radiotherapy	181
10	M	52										

**Table 2 ijms-22-08236-t002:** List of cancer associated proteins in platelets.

			Current Study		Sabrkhany et al.	
Gene Symbol	Protein Name	Entrez ID	Fold Change	*p*-Value	Fold Change	*p*-Value
CTSG	Cathepsin G	1511	3.69	0.025	2.89	0.012
PGLYRP1	Peptidoglycan recognition protein 1	8993	10.14	0.013	Only in cancer	0.005
RNF213	Ring finger protein 213	57674	1.81	0.043	1.70	0.005
RPL8	Ribosomal protein L8	6132	12.71	0.038	Only in cancer	0.024
S100A8	S100 calcium binding protein A8	6279	1.97	0.043	1.58	0.029
S100A9	S100 calcium binding protein A9	6280	1.90	0.034	1.66	0.007
GPX1	Glutathione peroxidase 1	2876	−1.38	0.008	−1.20	0.020
TNS1	Tensin 1	7145	−2.93	0.001	−1.59	0.035

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
