# Peer review of "Effects of Cancer Presence and Therapy on the Platelet Proteome"

_ijms, 2021, doi:10.3390/ijms22158236_

Round 1

Reviewer 1 Report

The article by Walraven et al, explores tha anlysis of the platelet proteome in cancer, as to identify novel biomarkers of disease.

In general, the article is well written and organized, while the reviewer noted that the manuscript requires still minor spell/style corrections.

While the study is of interest, the reviewer has noted some conceptual and methodological weknesses, that should be addressed.

  • Patients should be better characterized (cancer stage, etc, treatment, and specially referring to anti-tumor treatments, the authors should indicate which is which).
  • Platelets should be better and thoroughly characterized on every proteomic study. On the fresh blood sample and after washing, before preparing the protein extracts. As the authors comment in the introduction, some pathologies result on platelet activation in the circulation, and this will also affect their proteome (with consequences on the interpretation of data: i.e. are platelets produced differently, with a different quality signature at the proteome level, or are they altered in the circulation?). Furthermore, the protocol the authors present includes quite a high number of centrifugations, and even if they add PGI2, it is important to evaluate that platelets remain as close as possible to their phenotype in the fresh blood sample. To solve this, or add more information to this matter, it is important to show the CBC of the blood sample (fresh), and after all the centrifugations, prior protein lysate preparation, including especifically PLT count and MPV. Another important aspect, is to analyse their potential pre-activation status. This can be done by flow cytometry, measuring surface levels of CD62P before and after stimulation with a platelet agonist (fresh blood and after washing platelets). In this way, the authors could verify whether there is pre-activation or anergy.... because if platelets have degranulated (basally) in the circulation, or during the washing, this will also affect their proteome. --> since it is not possible to retrospectively analyse patient´s platelets (unless this has been done at the time), this analysis could be included from a couple of controls and cancer platelet samples.
  • The reviewer wonders why the authors use in-gel digestion instead of in solution digestion.
  • Regarding protein differential expression, the methodology is not explained in detail, which is rather surprising, being this a merely proteomics paper. The reviewer finds very important and relevant to provide with more information on methodology (some of the intrinsic questions are deduced after looking to the Supplementary tables, but one can not exspect to be right on all deductions... it should be described rigorously. There are issues such as missing values, normalization, imputation, low abundance (the authors mention that briefly), and bias intrinsic to the study, which has not been matched regarding age.
  • The conclusions are a bit beyond what the results really show. "Pan-cancer signature” might not be the right terminology for what has been done in the article, and can be highly misleading. Most of the cancers included in this study are cancers of the digestive tract, and as mentioned above, they are not appropriately described at the clinical level. If the authors aim at identifying biomarkers, the clinical relevance should be rigurously portrayed. Going back to the term "pan-cancer", this would be acceptable when 20/30 types of cancer are being analyzed, containing each a number of biological replicates (not one of each). 
  • The analysis before and after treatment is misleading as well, becuase it is done by grouping different types of cancer and treatment (not specified). Differences in platelet proteins are always identified mathematically and statistically, even comparing healthy controls of different ethnicities or age groups.
  • Please add a PCA or a similar tool to compare or analysis, including treated ones.

  • In line with this, there are four patients that are receiving acetylsalicylic acid. The medication could also be impacting platelet content. What was the criteria behind patient selection?

  • Lastly, since this analysis supports what was previously found by the authors (ref 28), it is surprising that a validation has not be performed on platelet samples from other patients (or even on these samples) by other techniques (western, ELISA,...), as to devlop the aimed biomarker identification.

Minor editing issues on Figures and Table presentations:

Figures and tables should be improved.

Figure 1, only the quality should be examined (image), to be applied to Figs. 3, 5 and 6.

Table 1: it should not be a copy-paste insert from Excel.

Figure 2: Please identify the sample on the heatmap, matching the samples on Table 1 (This should be applied to Figure 4 and Supplementary Fig 2).

Table 2, Check image quality (it is not sharp).

All supplementary Tables are too raw. Please, edit and explain, with more detailed legends

Reviewer 2 Report

The authors define as aim of their current study to identify a pan-cancer protein signature that could be used to discriminate cancer patients from healthy individuals, and to explore the effect of cancer therapy on the platelet proteome. They studied platelets from 9 patients with a variety of cancer types and 10 healthy controls. From the cancer patients, 3 were selected to analyse their platelet proteome before and after anti-tumor therapy. Using global protein profiling by mass spectrometry-based proteomics, they conclude that the platelet proteome is not only affected by cancer presence, but also by antitumor therapy. In a relevant preceding study (ref. 28 in the present paper), the authors showed that the platelet proteome of cancer patients is different from that of healthy individuals in early stages of the disease, and normalizes after surgical resection of the tumor, thus presenting a lucrative (???) source for potential cancer biomarkers [28].

The topic of this paper is clearly of considerable interest, and the authors have a very good track record in this field. However, there are a number of important points which need to be addressed by the authors.

  • Platelet isolation , protein digestion, quantitative analysis of nanoLC-MS/MS data.

It is  not entirely clear why the authors used NuPAGE gels to separate platelet proteins followed by nanoLC—MS/MS.  This procedure may result in some loss of certain proteins / protein groups. An alternative would be to digest the platelet lysates directly. They report that spectral counting was used to quantitative the protein abundance but only relative changes were reported. It would be desirable if the protein abundance is reported in biochemical terms such as fmol and/or copy number per platelet.         

  • They authors identified 4059 unique proteins overall, with an average of 2912 identified proteins in patient samples and 2808 in healthy control samples. They also detected 4384 non-redundant platelet proteins in a previous study (ref. 28 of their present paper.) However, the authors did no compare their platelet protein spectrum (quantitatively and/or qualitatively) to other data published in the field. It appears that there is a substantial loss of proteins compared to the full spectrum of platelet proteins obser4ved by others.
  • This paper reports the platelet proteome from 10 healthy donors (age 42-60, 1 female) and 9 tumor patients (age 50-79, 1 female) with various types of malignancies such as colon, rectal, anal, esophageal, gastric, tongue and primitive neuroectodermal cancer. From these cancer patients, 3 tumor patients (gastric ca, neuroectodermal ca, anal ca) provided platelet samples for proteomic analysis before and after tumor treatment It is clear that this patient cohort is very small (and also very heterogenous, as the authors also partially acknowledge in discussion). Also, patients (and probably also healthy volunteers, not reported) received various previous and present comedications (aspirin, plavix, diclofenac and others). This referee is quite aware of the difficulties to obtain appropriate controls.  However, I would be very careful with clinical conclusions and would avoid overstatements. For example, the title of this paper can be considered as overstatement, also sentences such as “we demonstrate that the platelet proteome of patients is affected by cancer presence and cancer therapy”.
  • Pan-cancer signature (Fig. 6/Table 2). The authors combined data from the current study with a previous  cancer platelet proteomics dataset [ 28], and report a pan-cancer proteome signature of 8 platelet proteins (RNF213, CTSG, PGLYRP1, RPL8, S100A8, S100A9, GPX1 and 196 TNS1) with more than 1.5 fold elevation ( 6 proteins) or more than 1.5 fold down-regulation ( GPX1, TNS1) ,  representing candidate cancer biomarkers within blood platelets.These are potentially very interesting findings, but they are far away from the criteria required for a biomarker.
  • Potential pan cancer marker. The authors suggest 8 up- (6) or down (2) -regulated proteins as pan cancer signature (see above). Were any attempts undertaken to validate the reported MS changes of these proteins by other methods (immunological assays, activities etc.).?)
  • Clinical data. If the authors combine clinical data from the present study and the previous one (ref.28) the most pertinent clinical data of all patients should be provided (perhaps as supplement table). This includes the general medication of both controls and patients.      

Round 2

Reviewer 1 Report

The revised manuscript has improved considerably.

Still the reviewer considers important to include a PCA, in addition to their preferred clustering.

See response: Please add a PCA or a similar tool to compare or analysis, including treated ones.
RESPONSE: Thank you for your comment. We have presented an unsupervised clustering of the data in the supplementary Figure 2, which is our preferred method to get an overall visualization of the data

Reviewer 2 Report

The authors provided an extensive response for this referee`s criticism. They also made important changes/ additions in the manuscript and provided now a thoroughly revised paper. There is one remaining  point which is still very critical. Considering the heterogeneity of their patient platelet samples and the small sample size altogether,  the diction of the title “Platelet proteome is affected by cancer presence and therapy”  is too strong, also the related conclusion “ In conclusion, the current study shows that the platelet proteome is affected in cancer patients. Furthermore, it utilized changes in the platelet proteome of patients to distill cancer associated proteins within platelets. This study also uncovers for the first time significant effects of antitumor therapy on the platelet proteome.”

This otherwise very interesting paper does not have the data to make such strong and decisive statements. They authors have modified their strongest clinical statements within the revised manuscript (i.e. addition of “may”, “potential “, future tasks etc.). This referee strongly suggests to the authors that they modify also their remaining overstatements, for example a small change in their title such as  “ Effects of cancer presence and therapy on the platelet proteome”. Such changes are also in the interest of the authors. In clinically  relevant research and especially cancer research it is mandatory to avoid over-statements.
